

# Online search trends and word-related emotional response during COVID-19 lockdown in Italy: a cross-sectional online study

Maria Montefinese[1,2,*], Ettore Ambrosini[1,3,4,*] and Alessandro Angrilli[1,4]

[1] Department of General Psychology, University of Padua, Padua, Italy
[2] Department of Experimental Psychology, University College London, University of London, London, United Kingdom
[3] Department of Neuroscience, University of Padua, Padua, Italy
[4] Padua Neuroscience Center, University of Padua, Padua, Italy
[*] These authors contributed equally to this work.

## ABSTRACT

**Background**. The strong and long lockdown adopted by the Italian government to limit COVID-19 spreading represents the first threat-related mass isolation in history that can be studied in depth by scientists to understand individuals' emotional response to a pandemic.

**Methods**. We investigated the effects on individuals' mental wellbeing of this long-term isolation by means of an online survey on 71 Italian volunteers. They completed the Positive and Negative Affect Schedule and Fear of COVID-19 Scale and judged valence, arousal, and dominance of words either related or unrelated to COVID-19, as identified by Google search trends.

**Results**. Emotional judgments changes from normative data varied depending on word type and individuals' emotional state, revealing early signals of individuals' mental distress to COVID-19 confinement. All individuals judged COVID-19-related words to be less positive and dominant. However, individuals with more negative feelings and COVID-19 fear also judged COVID-19-unrelated words to be less positive and dominant. Moreover, arousal ratings increased for all words among individuals with more negative feelings and COVID-19 fear but decreased among individuals with less negative feelings and COVID-19 fear.

**Discussion**. Our results show a rich picture of emotional reactions of Italians to tight and 2-month long confinement, identifying early signals of mental health distress. They are an alert to the need for intervention strategies and psychological assessment of individuals potentially needing mental health support following the COVID-19 situation.

Corresponding author
Ettore Ambrosini,
ettore.ambrosini@unipd.it,
ettore.ambrosini@gmail.com

## INTRODUCTION

Coronavirus disease 2019 (COVID-19) is a novel and emerging infectious disease caused by a new coronavirus strain named Severe Acute Respiratory Syndrome Coronavirus-2 (SARS-CoV-2) mainly transmitted by respiratory droplets and contact (*Sohrabi et al., 2020*; *Wu et al., 2020*). COVID-19 has quickly spread worldwide since December 2019, infecting millions of people and causing hundreds of thousands of deaths so that the World Health Organization (WHO) has announced the COVID-19 outbreak a pandemic. In order to cut the rate of new infections and flatten the COVID-19 contagion curve, health and political authorities imposed mass home-confinement directives and unprecedented severe restrictions on daily living. Italy, one of the worst-hit countries by the pandemic (at least in the first phase outside China), imposed a strict lockdown for over two months.

While social isolation and quarantine are imperative to abate the virus spread, the effects of these measures on the emotional wellbeing and mental health are just starting to be investigated. Indeed, individuals are reporting that the COVID-19 pandemic is increasing the levels of negative emotions and decreasing those of positive ones, contributing to a number of negative psychological, behavioral and health problems, such as, anxiety and depression (*Rossi et al., 2020*), abuse of alcohol and drugs, trouble in concentrating, increased aggressive behavior, maladaptive eating, and worse job performance (*Kirzinger et al., 2020*; *Smith, 2020*).

The perception of a pandemic threat through invasive media communication, such as that related to COVID-19, can induce fear-related emotions (*Van Bavel et al., 2020*). The dimension theory of emotions (*Osgood & Suci, 1955*) assumes that emotive space is defined along three dimensions: valence (indicating the way an individual judges a stimulus; from unpleasant to pleasant), arousal (indicating the degree of activation an individual feels towards a stimulus; from calm to excited) and dominance (indicating the degree of control an individual feels over a given stimulus; from out of control to in control). Fear is characterized as a negatively valenced emotion, accompanied by a high level of arousal (*Witte, 1992*; *Witte, 1998*) and a low dominance (*Stevenson, Mikel & James, 2007*). This is generally in line with previous results showing that participants judged stimuli related to the most feared medical conditions as the most negative, the most anxiety-provoking and the least controllable (*Warriner, Kuperman & Brysbaert, 2013*). Fear is also characterized by extreme levels of emotional avoidance of specific stimuli (*Perin et al., 2015*) and may be considered a unidirectional precursor to psychopathological responses within the current context (*Ahorsu et al., 2020*). Humans, indeed, possess a defensive system for fighting ecological threats (*LeDoux, 2012*; *Mobbs et al., 2015*). Previous studies have reported that fear-related emotions can lead individuals to engage in protective behaviors (*e.g.*, improving health knowledge) and often maladaptive behaviors (*e.g.*, stigmatization and discrimination) (*Rogers & Prentice-Dunn, 1997*; *Ruiter et al., 2014*; *Witte, Meyer & Martell, 2001*).

A meta-analysis reported that targeting fears can be valuable in some situations (*Witte & Allen, 2000*): when individuals believe they are able to defense themselves, strong fear can lead them to the adaptive danger control behavior; on the contrary, when individuals

feel helpless to act, strong fear can lead to maladaptive control actions such as defensive avoidance or reactance (*Van Bavel et al., 2020*; *Witte & Allen, 2000*). More importantly, dealing with fear in a pandemic situation could be easier for some people than others. Indeed, individual differences have been associated with behavioral responses to the pandemic status (*Carvalho Pianowski & Gonçalves, 2020*).

To mitigate the COVID-19 effects on individuals' mental health, it is compelling to evaluate their emotional response to this emergency. Internet searches is a direct tool to address this issue. Indeed, it has been reported that COVID-19 affected the content that people explored online (*Effenberger et al., 2020*), and online media and platforms offer essential channels where people convey their feelings and emotions and seek health-related information (*Kalichman et al., 2003*; *Reeves, 2001*). In particular, Google Trends is an available data source of real-time internet search pattern, which has been demonstrated to be a valid indicator of people's desires and intentions (*Payne, Brown-Iannuzzi & Hannay, 2017*; *Pelham et al., 2018*). Thus, the amounts of COVID-19-related internet searches revealed by Google Trends are an indicator of how people feel about concepts related to the COVID-19 pandemic. A shift in online search trends reflects a change in participants' interests and attitudes towards a specific topic. Based on the topic, the context (*i.e.,* the reasons causing this change), and this mutated interest *per se*, it is possible to predict people's behavior and affective response towards the topic in question.

In this study, we aim to understand how emotional reaction and online search behavior has changed in response to the COVID-19 lockdown in the Italian population. Studying the emotional response of Italians is important because Italy was the first Western country to experience a large number of COVID-19 cases and to adopt the strongest national lockdown for over two months (it started on March 10th, one day before the WHO has announced the COVID-19 pandemic status, and ended on May 18th) (*Stella, Restocchi & De Deyne, 2020*). In this regard, we expect that an ongoing pandemic threat to the individuals' health may elicit a change in the online behavior and emotional reactions, especially for individuals that feel the current situation with more fear and negative emotional state.

Findings might inform about the real-time estimation of the COVID-19 pandemic impact on participants' emotional response and will provide accurate insights on the mental wellbeing of the population. This new knowledge could provide some guidelines for more punctual intervention strategies for individuals in need of mental health support following the COVID-19 situation.

## MATERIALS & METHODS

We report how we determined our sample size, all data exclusions, all inclusion/exclusion criteria, all manipulations, and all measures in the study. All inclusion/exclusion criteria were established prior to data analysis. All data and materials are available from our project repository on the Open Science Framework (https://osf.io/32xab). No part of the study, including the analyses, was pre-registered.

## Selection of experimental stimuli

We used Google Trends (https://trends.google.com/trends/) to assess internet activity related to the COVID-19 epidemic in Italy in the first four months of 2019 and 2020. The period before Italy's first confirmed COVID-19 patient (February 21st, 2020) was included as a baseline to assess the COVID-related change in the temporal pattern of online searches. Indeed, Google Trends determines the normalized proportion of searches for user-specified terms among all searches performed using Google for a given location and time period, expressed as the relative search volume (RSV) with a datapoint for each day, scaled on a [0, 100] range where 100 is the maximum search interest for the time and location selected. Moreover, data from 2019 were used to control for potential unspecific seasonal trends or idiosyncratic temporal patterns in RSV data (for example, the word "freedom" -*libertà* in Italian- shows a peak on April 25th, the Liberation Day in Italy).

The following terms were used: "coronavirus", "COVID", "COVID-19", and "virus". We also extracted the RSV for the 1121 words included in the Italian adaptation of the English affective norms (ANEW; *Montefinese et al., 2014*) by using the gtrends R package (*Massicotte & Eddelbuettel, 2016*) for R (*R Core Team, 2019*). RSV data for one word (mildew) were not available. We retrieved RSV data from January 1st to April 27th (most current data available at the time of data retrieval), for both 2019 and 2020 years.

The experimental stimuli were selected among the Italian ANEW words by assessing to what degree the temporal dynamics in their search trends was specifically related to that of the search trend for the COVID-19 terms. We aimed to identify the Italian ANEW words that consistently showed the greater change in internet activity due to the COVID-19 epidemic, while controlling for unspecific RSV trends. This was done by taking four different analytical approaches based on a multiverse analysis (*Steegen et al., 2016*).

First, for each year, a COVID-related RSV time series (COVID-RSV) was computed by averaging the RSV time series for the four COVID-related terms. Pearson's correlation coefficients ($r$) were then computed between the COVID-RSV and those for the Italian ANEW words (ANEW-RSV). These $r$ values thus reflect the strength of the association between the COVID-RSV and each ANEW-RSVs for both the 2019 and 2020. Next, we compared the $r$ values for 2020 and 2019 by performing Steiger's $Z$ tests for non-overlapping correlations based on dependent groups (*Steiger, 1980*), thus obtaining a $Z$ value $Z_{Pears}$ for each ANEW word. Second, we computed differential RSV time series by subtracting the COVID-RSV and ANEW-RSVs for 2019 from those for 2020 and computed their Pearson's correlation coefficients ($r_{diff}$). Both $Z$ and $r_{diff}$ values reflect the 2020-specific change in the strength of the association between the COVID-RSV and each ANEW-RSVs.

The same procedure described in the previous paragraph was performed after rank-transformation of all original RSV data to compute non-parametric Spearman's correlation coefficients $\rho$ and $\rho_{diff}$, as well as the $Z_{Spear}$ value from the Steiger's tests comparing $\rho$ values for 2020 and 2019. This was done to control for both non-normality of our data and potential outlier observations. We thus obtained four differential correlation measures ($r_{diff}$, $\rho_{diff}$, $Z_{Pears}$, and $Z_{Spear}$) reflecting the (signed) degree of the specific impact of the COVID-related interest on the search trends for each ANEW word.

Based on these correlational measures, we selected three groups of stimuli, each composed by 20 words, as described below. This number of stimuli was the largest that can be reliably rated by each participant during a single online session in a reasonable amount of time (based on pilot testing and Montefinese et al.'s normative study (2014), which used 56–57 stimuli for each participant), ensuring the reliability of the ratings and yielding the maximum possible power. The first group (REL+) consisted in the words showing the largest positive relation between their search trends and the search trend for the COVID-related terms. By contrast, the second group (REL-) consisted in the words showing the largest negative relation between their search trends and the search trend for the COVID-related terms. In other words, the COVID-19 epidemic in Italy, and the consequent increase in interest for the COVID-related terms, was related to a similar increase of interest for the REL+ words and a decrease of interest for the REL- words. The third group (UNREL) consisted in the words for which the search trend was unrelated to the search trend for the COVID-related terms.

The REL+ and REL- words were selected as those consistently showing, respectively, the highest and the lowest $r_{diff}$, $\rho_{diff}$, $Z_{Pears}$, and $Z_{Spear}$ values. Specifically, we first selected the words that were in the top (or bottom, respectively) 2.5% of the distribution for at least three out of the four differential correlation values, and then selected the words with the largest differential correlation values that were in the top (or bottom, respectively) 2.5% of the distribution for at least two out of the four differential correlation values. The UNREL words were selected as those showing the smallest differential correlation values. For all the three groups, the selection was limited to nouns and verbs.

Figure 1 shows the differential RSV time series for the COVID-related terms and one exemplar stimuli for each of the REL+ (fever, *febbre* in Italian), REL- (hotel), and UNREL ([to] disturb, *disturbare* in Italian) groups. This figure illustrates the clear COVID-related increase of online searches for the REL+ word "fever", likely due to health concerns, as well as the clear COVID-related decrease of online searches for the REL- word "hotel", likely due to a more limited mobility suddenly imposed to Italians during the COVID-19 lockdown.

The selected experimental stimuli are available at Open Science Framework (see https://osf.io/2mc3k/, Table S1).

## Procedure

An online survey was conducted using Google Forms (https://www.google.com/forms/about/) to collect affective ratings during the lockdown caused by the COVID-19 epidemic in Italy. The first section of the form consisted of the informed consent, including a basic description of the study, followed by a section asking participants to specify their gender, age, and education level. The next sections of the form consisted in the Positive and Negative Affect Schedule (PANAS, *Terraciano, McCrae & Costa, 2003*) and Fear of COVID-19 Scale (FCV-19S, *Ahorsu et al., 2020*) questionnaires to evaluate participants' positive and negative affective state (assessed, respectively, with the PANAS+ and PANAS- subscales of the PANAS) and fear of COVID-19, which we expected to modulate participants' affective ratings. Finally, in the last section of the form participants were asked to provide their

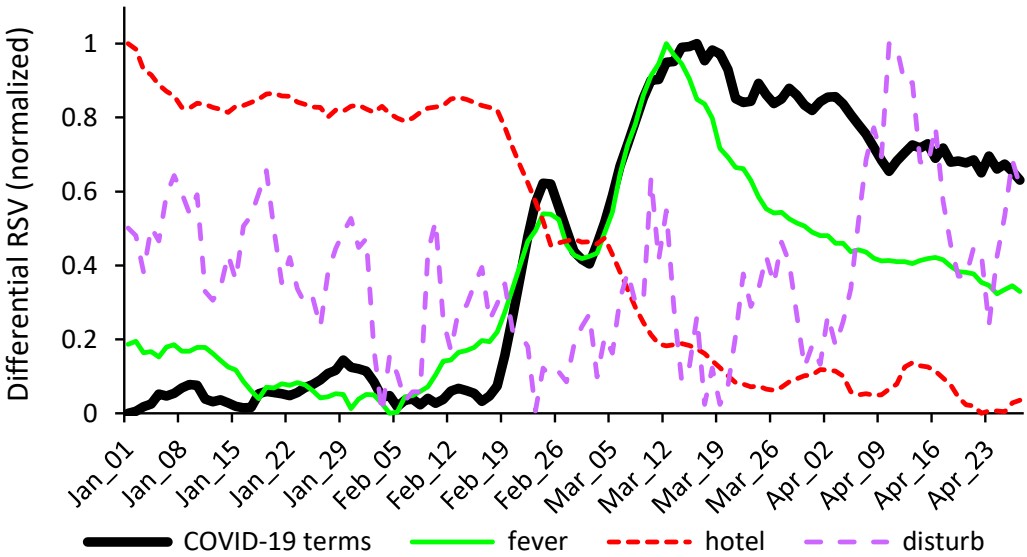

**Figure 1** **Search trends for exemplar experimental stimuli.** The line plots represent the differential RSV time series for the COVID-related terms (black solid line) and for an exemplar stimulus for each group of words selected: fever (green solid line, for the REL+ group) and hotel (red dashed line, for the REL- group), which showed respectively the largest positive and negative correlation with the data for the COVID-related terms, and disturb (purple dotted line, for the UNREL group), which showed the smallest absolute correlation with the data for the COVID-related terms. The differential RSV time series were normalized in the [0, 1] range for visualization purposes.

affective ratings for the 60 experimental stimuli, which were presented in a random order. Specifically, participants were instructed to rate how they felt when reading each word along the three affective dimensions of valence, arousal, and dominance by using the 9-point self-assessment manikin (*Lang, 1980*). The format and instructions for the affective rating task were the same as those used in our previous work (*Montefinese et al., 2014*).

Data were collected in the period from May 4th to May 17th, 2020, the last day of full lockdown in Italy, from 71 adult native Italian speakers (56 females and 13 males; mean (SD) age = 26.2 (7.9) years; mean (SD) education = 15.3 (3.2) years). There were no other specific eligibility criteria. Participants consisted of a convenience sample recruited *via* online advertisements through social networks or identified *via* researchers' personal networks. It is important to note that in the present study affective ratings were provided for each word by twice as many participants, on average, as compared not only to the Italian ANEW norms (*Montefinese et al., 2014*), but also to affective norms in general (*e.g.*, *Warriner, Kuperman & Brysbaert, 2013*), so assuring an adequate reliability and generalizability of our affective ratings. It is also important to note that most of our research questions involved by-items statistical analyses, so the number of participants did not directly impact on the statistical power of our analysis.

A first sensitivity power analysis was conducted in G\*Power for a mixed ANOVA on current and normative ratings with three groups of 20 words, assuming a correlation between repeated measures of .80 (as estimated conservatively from our previous study,

*Montefinese et al., 2014*). This analysis revealed that our sample size (60 words) was large enough to detect a small effect size (Cohen's $d = .12$, corresponding to $\eta^2_p = .014$) with a power of .80. We also used the method introduced by Westfall and colleagues (*2014*) to perform a sensitivity power analysis for a stimuli-within-condition linear mixed-effects model, assuming participants, stimuli, and residual variance partitioning coefficients of .1, .15, and .75, respectively (as estimated conservatively from some recent unpublished studies with a similar design from our research group). This analysis revealed that our sample size (71 participants and 60 words) was large enough to detect a small-medium effect size ($d = .30$) with a power of .80.

The procedure used in the study is in accordance with the ethical standards of the 2013 Declaration of Helsinki for human studies of the World Medical Association. The project has been approved by the Ethical Committee for the Psychological Research of the University of Padova (approved protocol reference number: 3563).

## Data analysis

We performed a series of analysis to investigate (1) the relation between the lexical and affective variables for the words we used and the COVID-dependent changes in their online searches; (2) the reliability of the present affective ratings; (3) the impact of the COVID-19 lockdown in Italy on affective ratings; (4) the effect of participants' emotional profile on affective ratings.

A first set of analyses was conducted to investigate whether the magnitude of the specific impact of the COVID-related interest on the search trends for Italian ANEW words could be explained by their lexical and affective variables taken from Italian ANEW norms (*Montefinese et al., 2014*). To this aim, we first computed the zero-order parametric and non-parametric correlations between the four differential correlation measures ($r_{diff}$, $\rho_{diff}$, $Z_{Pears}$, and $Z_{Spear}$), on the one side, and the valence, arousal, dominance, familiarity, concreteness, and word frequency, on the other side. The differential correlation values were first transformed to improve the normality of their distribution by performing a natural log-transformation on their absolute values. We also performed stepwise multiple regression analyses with each transformed differential correlation value as the dependent variable and the lexical and affective variables as predictors. We used a tolerance cutoff of .6 to minimize multicollinearity and maximize precision of the regression parameter estimates.

We also assessed the reliability of the present affective ratings by correlating them with those of the Italian ANEW norms (*Montefinese et al., 2014*) and by computing split-half correlations corrected with the Spearman-Brown formula after 10,000 randomizations.

We then investigated the impact of the lockdown imposed by the COVID-19 epidemic on affective ratings. To this aim, we compared the affective ratings collected in the present sample with those collected in the normative sample for the same stimuli. First, for each affective dimension, we performed a two-tailed paired $t$-test contrasting the mean ratings from the present and normative samples; we also performed two-tailed Welch's $t$-tests contrasting the individual ratings for each word, followed by an internal meta-analysis to estimate combined effect sizes. Moreover, we investigated whether the lockdown-dependent

| Table 1 | Descriptive statistics. | | | | | | | | | | | |

| | | | | Affective ratings | | | | | | | | |
| | | | | REL- | | | REL+ | | | UNREL | | |
| | PANAS- | PANAS+ | FCV-19S | VAL | ARO | DOM | VAL | ARO | DOM | VAL | ARO | DOM |
|---|---|---|---|---|---|---|---|---|---|---|---|---|
| *Present sample (n = 71)* | | | | | | | | | | | | |
| **M** | 19.99 | 28.08 | 13.56 | 5.80 | 5.17 | 4.98 | 5.58 | 5.47 | 4.88 | 4.06 | 5.63 | 4.26 |
| **SD** | 7.47 | 6.68 | 5.46 | 0.58 | 0.76 | 0.55 | 0.49 | 0.85 | 0.62 | 0.48 | 0.66 | 0.62 |
| **min** | 10 | 16 | 7 | 4.45 | 3.35 | 3.75 | 4.45 | 3.5 | 3.55 | 3.05 | 4.20 | 2.95 |
| **max** | 37 | 48 | 29 | 7.05 | 6.65 | 6.30 | 6.90 | 7.10 | 6.40 | 5.20 | 6.80 | 5.50 |
| *Normative data (mean n = 34.5)[a]* | | | | | | | | | | | | |
| **M** | | | | 6.52 | 5.18 | 5.82 | 6.33 | 5.65 | 5.44 | 4.45 | 6.10 | 4.76 |
| **SD** | | | | 1.56 | 0.92 | 0.72 | 2.04 | 0.85 | 1.11 | 2.21 | 0.87 | 1.04 |
| **min** | | | | 1.72 | 2.82 | 4.76 | 2.25 | 4.06 | 3.30 | 1.79 | 4.83 | 2.76 |
| **max** | | | | 8.67 | 6.97 | 7.50 | 8.56 | 7.39 | 7.18 | 8.24 | 7.88 | 7.09 |

**Notes.**

[a] Data computed from the Italian ANEW norms (*Montefinese et al., 2014*).

PANAS-, negative subscale of the PANAS; PANAS+, positive subscale of the PANAS; FCV-19S, fear of COVID-19 scale; VAL, valence; ARO, arousal; DOM, dominance.

differences in affective ratings were modulated by the specific impact of the COVID-related interest on the search trends for our stimuli. For each affective dimension, we performed a by-items Welch's ANOVA on the raw difference in the affective ratings between the present and the normative samples, with Stimulus Type (REL+, REL-, UNREL) as a between-items factor. Post-hoc comparisons were performed using Welch's $t$ tests.

Lastly, we investigated whether participants' affective state and fear of COVID-19 modulated their affective ratings. To this aim, for each affective dimension we performed three linear mixed-effects model (LMM) analyses with the raw difference in the affective ratings as the dependent variable, three parameters for (1) the fixed effects of Stimulus Type, (2) either the PANAS-, PANAS+, or FCV-19S (centered), and (3) their interaction, and by-subjects and by-items random intercepts.

# RESULTS

All data and materials necessary to replicate our analyses are available on the Open Science Framework (https://osf.io/32xab), including participants' demographic variables and scores for the FCV-19S and the PANAS- and PANAS+ subscales of the PANAS (Supplemental Material available online at https://osf.io/2mc3k, Table S2), as well as a STROBE checklist (Table S6). Table 1 shows the descriptive statistics for the scales and affective ratings collected in the present study, as well as for the affective ratings from the normative study (Italian ANEW norms, *Montefinese et al., 2014*) for the same words we used here.

All the results were very similar across the four differential correlation measures we used, suggesting that deviations from normality and potential outliers did not bias substantially our results. For the sake of brevity, we report here the results for the $Z_{Spear}$ measure, which assures the greatest protection against potential biases.

### Impact of lexical and affective variables on COVID-dependent changes of ANEW search trends

All the correlations were significant ($p < .001$), but (at best) moderate in size (for all the results, see Table S3, https://osf.io/2mc3k).

The final model for the multiple regression analysis included four predictors ($F$ (4, 1108) $= 52.6$, $p < .001$, $R^2 = 15.95\%$; seven cases were not included due to missing word frequency data; see Supplemental material, https://osf.io/8hpek). Results showed that the specific impact of the COVID-related interest on the search trends was greater for the Italian ANEW words with higher word frequency ($b = 0.067$, 95% confidence interval (95% CI) $= [0.050–0.085]$; $t = 7.46$; $p < .001$), concreteness ($b = 0.095$, 95% CI $= [0.073–0.012]$; $t = 8.48$; $p < .001$), and valence ($b = 0.041$, 95% CI $= [0.023–0.059]$; $t = 4.53$; $p < .001$), as well as for the Italian ANEW words with lower arousal ($b = -0.054$, 95% CI $= [-0.095, -0.012]$; $t = -2.52$; $p = .012$).

### Reliability analysis on affective ratings

The reliability analysis showed very high correlations between the Italian ANEW norms (*Montefinese et al., 2014*) and the affective ratings collected in the present sample, especially for the valence (.98, .81, and .79 for valence, arousal, and dominance, respectively), and the median split-half correlations were even higher (.99, .93, and .97, for valence, arousal, and dominance, respectively; range $= [.97, .99]$, $[.74, .97]$, and $[.93, .99]$).

### Lockdown impact on affective ratings

The analyses revealed that the lockdown imposed by the COVID-19 epidemic affected participants' affective ratings. Indeed, as compared to the normative sample, our participants rated the experimental stimuli with lower valence (mean difference $M_{diff} = -0.625$, 95% CI $= [-0.746, -0.503]$; $t$ (59) $= -10.27$; $p < .001$; $d = -1.325$, 95% CI $= [-1.670, -0.974]$), arousal ($M_{diff} = -0.220$, 95% CI $= [-0.363, -0.077]$; $t$ (59) $= -3.08$; $p = .003$; $d = -0.397$, 95% CI $= [-0.659, -0.133]$), and dominance ($M_{diff} = -0.635$, 95% CI $= [-0.808, -0.461]$; $t$ (59) $= -7.32$; $p < .001$; $d = -0.945$, 95% CI $= [-1.247, -0.638]$). These results were confirmed by the Welch's $t$-tests performed on each word, which revealed significant differences for 26, 7, and 27 words (corresponding to 43.33%, 11.67%, and 45% of the words) for valence, arousal and dominance, respectively (https://osf.io/2mc3k, Table S4; see also Fig. S1), as also suggested by the results of the internal meta-analysis. Most of these significant differences reflected lower affective ratings in the current sample, with an apparent difference in their distribution across the three types of stimuli (REL+, REL-, UNREL; see https://osf.io/2mc3k, Fig. S1). Indeed, for the valence, the combined effect sizes $d$ for REL+, REL-, and UNREL words were, respectively, $-0.340$ (95% CI $= [-0.403, -0.277]$), $-0.354$ (95% CI $= [-0.471, -0.237]$), and $-0.178$ (95% CI $= [-0.265, -0.092]$), with a significant difference across Stimulus Types ($Q^*(2) = 7.93$, $p = .019$). For the arousal, the combined effect sizes $d$ for REL+, REL-, and UNREL words were, respectively, .001 (95% CI $= [-0.068, .069]$), $-0.063$ (95% CI $= [-0.159, 0.034]$), and $-0.176$ (95% CI $= [-0.248, -0.105]$), with a significant difference across Stimulus Types ($Q^*(2) = 7.37$, $p = .025$). For the dominance, the combined effect sizes $d$ for

**Table 2  Results of the Welch's ANOVAs on rating differences and related descriptive statistics.**

| | Welch's ANOVA | | | | | REL+ | | REL- | | UNREL | |
|---|---|---|---|---|---|---|---|---|---|---|---|
| | $F$ | df1 | df2 | $p$ | $\eta^2_p$ | M | SD | M | SD | M | SD |
| Valence | 4.31 | 2 | 36.67 | 0.021 | 0.190 | −0.76 | 0.57 | −0.72 | 0.34 | −0.40 | 0.41 |
| Arousal | 4.80 | 2 | 37.22 | 0.014 | 0.205 | −0.18 | 0.64 | −0.01 | 0.49 | −0.46 | 0.44 |
| Dominance | 1.52 | 2 | 37.41 | 0.232 | 0.075 | −0.56 | 0.72 | −0.85 | 0.71 | −0.50 | 0.56 |

REL+, REL-, and UNREL words were, respectively, −0.333 (95% CI = [−0.445, −0.221]), −0.205 (95% CI = [−0.330, −0.079]), and −0.175 (95% CI = [−0.267, −0.083]), with no significant difference across Stimulus Types ($Q^\star(2) = 4.82$, $p = .090$).

These results were confirmed by the Welch's ANOVAs (see Table 2; see also https://osf.io/prx4s). Indeed, the COVID-related decrease in valence was significantly different across Stimulus Types, with a smaller decrease for UNREL words as compared to both REL- ($t$ (36.7) = −2.71; $p = .010$; $d = −0.858$, 95% CI = [−1.525, −0.172]) and REL+ ($t$ (34.8) = −2.30; $p = .028$; $d = −0.723$, 95% CI = [−1.378, −0.057]) ones, which in turn did not differ between each other ($t$ (31.2) = 0.23; $p = .818$; $d = .073$, 95% CI = [−0.548, 0.693]). Moreover, the COVID-related decrease in arousal was significantly different across Stimulus Types, but this time with a significantly larger decrease for UNREL words as compared to REL- ($t$ (37.6) = 3.09; $p = .004$; $d = 0.977$, 95% CI = [0.274, 1.166]), but not REL+ words ($t$ (33.6) = 1.61; $p = .116$; $d = 0.510$, 95% CI = [−0.137, 1.144]) ones, and no significant differences between REL- and REL+ words ($t$ (35.5) = 0.96; $p = .342$; $d = 0.305$, 95% CI = [−0.327, 0.928]). Finally, the COVID-related decrease in dominance did not significantly differ across Stimulus Types, with similar decreases (all $|t|$s < 1.72; $p$ s > .096; $|d|$s < 0.541).

To sum up the results of these analysis, they provided converging evidence revealing COVID-dependent changes of affective ratings, with lower valence especially for REL- and REL+ words, lower arousal especially for UNREL words, and lower dominance regardless of the word group.

### Effect of participants' emotional profile on affective ratings

The results of LMM analyses for the three affective dimensions are shown in Table 3 (see also Table S5, https://osf.io/2mc3k).

For the valence, the LMM analyses (see https://osf.io/hbnuc) confirmed that the decrease in valence ratings was significantly different across Stimulus Types and revealed that this effect was modulated by participants' PANAS- and FCV-19S scores. Indeed, the decrease in valence ratings was larger for participants with higher PANAS- scores ($F(1, 69) = 17.51$, $p < .001$) and this effect was significantly modulated by Stimulus Type ($F(2, 4128) = 6.42$, $p = .002$): the impact of PANAS- on the decrease in valence was smaller for both REL- and REL+ words, for which the decrease in valence was evident also for participants' with lower PANAS- scores; by contrast, the decrease in valence for UNREL words was evident in participants with higher PANAS- scores only (Fig. 2A). A similar pattern was observed for the model assessing the impact of participants' FCV-19S scores, with a significant two-way interaction ($F(2, 4128) = 15.06$, $p < .001$) as shown in Fig. 2C.

**Table 3  Results of the LMM analyses, omnibus tests for fixed effects.**

| Model (Effect) | Valence | | | | Arousal | | | | Dominance | | | |
|---|---|---|---|---|---|---|---|---|---|---|---|---|
| | *F* | df1 | df2 | *p* | *F* | df1 | df2 | *p* | *F* | df1 | df2 | *p* |
| **PANAS-** | | | | | | | | | | | | |
| StimType | 3.89 | 2 | 57 | .026 | 3.73 | 2 | 57 | .030 | 1.53 | 2 | 57 | .224 |
| PANAS- | 17.51 | 1 | 69 | <.001 | 12.32 | 1 | 69 | <.001 | 10.72 | 1 | 69 | .002 |
| StimType*PANAS- | 6.42 | 2 | 4128 | .002 | 1.44 | 2 | 4128 | .238 | 7.45 | 2 | 4128 | <.001 |
| **PANAS+** | | | | | | | | | | | | |
| StimType | 3.89 | 2 | 57 | .026 | 3.73 | 2 | 57 | .030 | 1.53 | 2 | 57 | .224 |
| PANAS+ | 0.68 | 1 | 69 | .413 | 4.40 | 1 | 69 | .040 | 10.98 | 1 | 69 | .001 |
| StimType*PANAS+ | 0.13 | 2 | 4128 | .877 | 3.86 | 2 | 4128 | .021 | 1.05 | 2 | 4128 | .351 |
| **FCV-19S** | | | | | | | | | | | | |
| StimType | 3.89 | 2 | 57 | .026 | 3.73 | 2 | 57 | .030 | 1.53 | 2 | 57 | .224 |
| FCV-19S | 1.03 | 1 | 69 | .314 | 5.40 | 1 | 69 | .023 | 7.07 | 1 | 69 | .010 |
| StimType*FCV-19S | 15.06 | 2 | 4128 | <.001 | 3.20 | 2 | 4128 | .041 | 2.26 | 2 | 4128 | .105 |

**Notes.**

StimType, stimulus type; df, degrees of freedom.

For the arousal, the LMM analyses (see https://osf.io/j4kym) revealed that participants' ratings were positively related to both their PANAS- ($F(1, 69) = 12.32$, $p < .001$; Fig. 2D) and FCV-19S ($F(1, 69) = 5.40$, $p = .023$) scores, and this latter effect was significantly modulated by Stimulus Type ($F(2, 4128) = 3.20$, $p = .041$): the impact of FCV-19S on arousal ratings was larger for both REL- and REL+ words, for which participants with higher FCV-19S scores tended to show an increase in arousal ratings, as compared to UNREL ones (Fig. 2F). By contrast, participants' arousal ratings were negatively related to their PANAS+ scores ($F(1, 69) = 4.40$, $p = .040$), especially for the REL- words as compared to the UNREL ones ($F(2, 4128) = 3.86$, $p = .021$; Fig. 2E).

For the dominance, the LMM analyses (see https://osf.io/5w7pc) revealed that participants' ratings were related positively to their PANAS+ scores ($F(1, 69) = 10.98$, $p < .001$; Fig. 2H), but negatively to both their FCV-19S ($F(1, 69) = 7.07$, $p = .010$; Fig. 2I) and PANAS- ($F(1, 69) = 10.72$, $p = .002$) scores, and this latter effect was significantly modulated by Stimulus Types ($F(2, 4128) = 7.45$, $p < .001$): the impact of PANAS- on the decrease in dominance was smaller for both REL- and REL+ words, for which the decrease in dominance was evident also for participants' with lower PANAS- scores; by contrast, the decrease in dominance for UNREL words was evident in participants with higher PANAS- scores only (Figs. 2G–2I). A similar pattern was observed for the model assessing the impact of participants' FCV-19S scores, as shown in Fig. 2I, but the two-way interaction did not reach the significance level ($F(2, 4128) = 2.26$, $p = .105$).

## DISCUSSION

The present study exploited Google Trends data to understand how online search behavior and emotional reactions to common concepts have changed in response to the COVID-19 lockdown in the Italian population. First, we found that the concepts more often searched online by the individuals during the lockdown were those with a higher frequency of use,

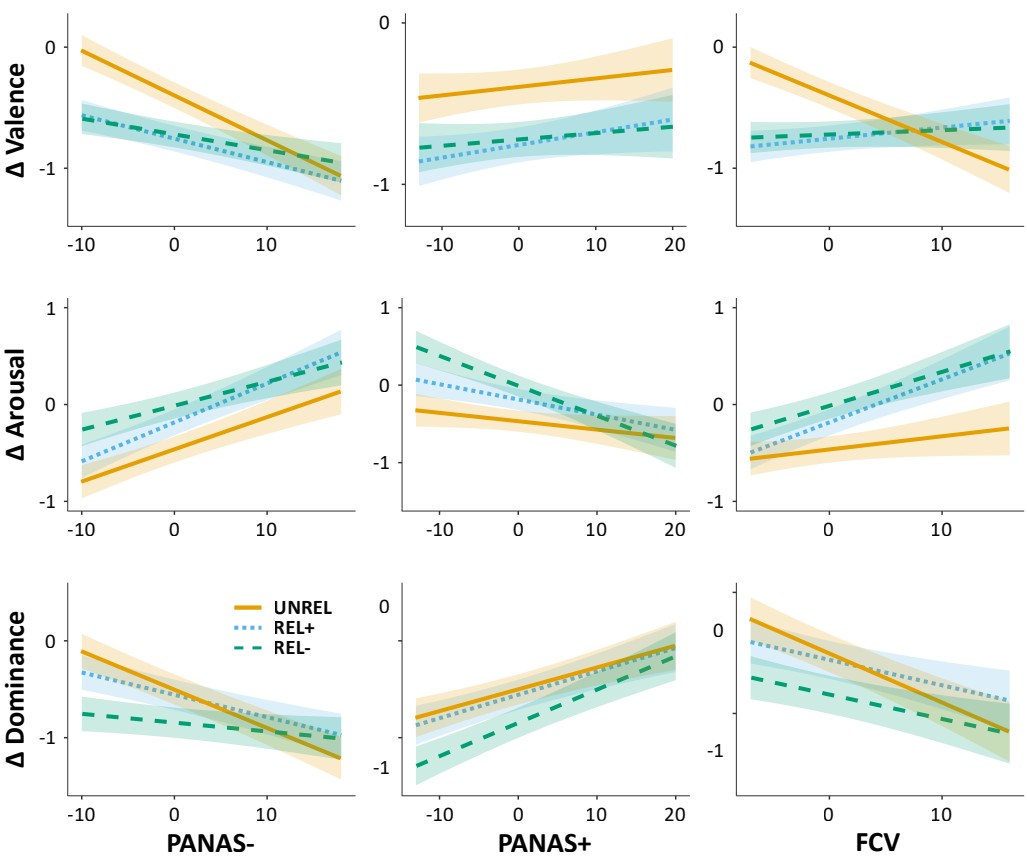

**Figure 2  Results of the LMM analyses, two-way interactions.** The line plots show the COVID-related differences (Δ) in affective ratings (Valence, top row; Arousal, middle row; Dominance, bottom row) as a function of both Stimulus Type (REL-, green dashed line; REL+, light blue dotted line; UNREL, orange solid line) and participants' affective state as measured by the PANAS- (left column), PANAS+ (middle column), and FCV-19S (FCV, right column) scores. The shaded regions represent the standard error of the mean.

those more concrete and positive, as well as those less arousing. These results suggest that intrinsic lexical-semantic properties per se were related to the COVID-related lockdown effect on individuals' online search interest.

We also asked participants to evaluate valence, arousal, and dominance of concepts (represented by Italian words) using the Self-Assessment Manikin (SAM) in a Web survey procedure. This type of approach informs on the relation between current context and individuals' emotions and mental distress, mostly from the perspective that emotions of the isolated individuals are conveyed mainly in the linguistic modality. Participants' ratings resulted highly reliable, especially the valence, corroborating previous findings (*Warriner, Kuperman & Brysbaert, 2013*; *Montefinese et al., 2014*). Indeed, the concept of valence is more straightforward since it is founded on ancestral motivational brain circuits that developed to ensure individual survival by reacting to appetitive and aversive environmental cues (*Lang & Bradley, 2010*). Accordingly, it has been shown that the valence dimension exists in all cultures (*Russell, 1991*).

 

Interestingly, we found that lockdown imposed by the COVID-19 epidemic had a substantial impact on participants' emotional responses, with lower affective judgments compared to the normative sample, especially for valence and dominance. In other words, when facing common concepts during COVID-related confinement, individuals experienced more negative feelings as well as feelings of being less aroused and less in control.

The results concerning the valence and dominance dimensions are consistent with the expected individuals' stronger feelings of fear and reduced sense of agency (and a consequent subjective perception of being in an out-of-control situation) in the current context, that is, an imminent threat to the humanity health (*Stevenson, Mikel & James, 2007*; *Warriner, Kuperman & Brysbaert, 2013*). Feelings of fear and reduced sense of agency might be the source of similar results found on previous studies using semantic and emotional network analysis on social discourse in Italian tweets at the end of the first lockdown (*Stella, Restocchi & De Deyne, 2020*; *Stella, 2020*). *Stella (2020)* showed that Italian participants tended to re-share a greater number of messages expressing fearful ideas, probably triggered by the strong affinity of the tweets' content and the feeling of individuals following the sudden raises in the COVID-19 contagion curve after the reopening. Fear is also the emotional concept most frequently produced by Italian participants in relation to the COVID-19 concept in a word association task (*i.e.,* participants listed concepts coming in mind in response to a given concept) (*Mazzuca et al., 2021*).

However, the result reported on the arousal dimension was quite unexpected, as the lockdown was expected to make individuals more activated in general. This apparent counterintuitive result was better qualified when considering the stimulus type in the analysis. The pattern of results was indeed driven by a decrease of arousal in participants for concepts unrelated to the COVID-19 topic (*e.g.*, orgasm, ocean), reflecting a loss of interest and activation in COVID-unrelated topics during the COVID-19 pandemic. A semantic network analysis of tweets posted in relation to the COVID-19 pandemic during a period of social restrictions found psychophysiological numbing in individuals across 19 countries: Twitter users increasingly fixate on mortality, but in a decreasingly emotional and increasingly analytic tone (*Dyer & Kolic, 2020*) Importantly, our results indicate that the individuals' subjective emotional profile modulated their lockdown-related changes in affective judgements of COVID-related and -unrelated concepts. Indeed, participants that felt the ongoing situation with less fear and a less negative affective state tended to rate only the COVID-related concepts with less valence and dominance, and all the concepts with less arousal. Concepts related to most feared medical conditions are also the most negative, the least controllable, and the most anxiety provoking (*Warriner, Kuperman & Brysbaert, 2013*), thus the affective reaction of these participants is understandable, also considering the limitations imposed by the lockdown. Moreover, this affective reaction could even be considered as somewhat adaptive, as it may promote the engagement in social distancing and restrictive behavior and, thus, the avoidance of situations that increase the risk of contagion.

Conversely, the participants with a more negative affective state presented the same pattern (*i.e.,* less valence and dominance) for the COVID-unrelated concepts as well,

but they were also more aroused by all the concepts. Their affective response was thus unspecific and potentially maladaptive (*Ruiter et al., 2014*; *Witte, Meyer & Martell, 2001*). Other studies have shown that negative effects of epidemic crisis and threat to the humanity such as higher anxiety and lower wellbeing affected individuals' mental health (*Kachanoff, Kapsaskis & Gray, 2020*; *Duncan, Schaller & Park, 2009*; *Pappas et al., 2009*). By means of network analysis, *Stella, Restocchi & De Deyne (2020)* detected emotions of anger, fear, and anxiety through social media in the Italian population following social distancing. When testing the effects of fear induction through film clips or virtual reality experience on participants' emotional reactivity, several studies revealed that in fear and threat conditions participants reported feeling less in control in combination with more arousal and negative valence (*Fernández-Aguilar et al., 2020*; *Palomba et al., 2000*; *Thomson et al., 2019*).

## CONCLUSIONS

Our results comprise initial evidence on the association between personality traits and social distancing during the COVID-19 pandemic. They show a rich picture of emotional reactions of Italians to a tight and 2-month long confinement, identifying early signals of mental health distress. Taken together with early surveys carried out on Italian samples on emotional response to COVID-19 pandemic (*Bischetti, Canal & Bambini, 2020*; *Rossi et al., 2020*), they are an alert to the need for intervention strategies and psychological assessment of individuals potentially needing mental health support following the COVID-19 situation. While online surveys and questionnaires may directly address this issue, they are limited by the difficulty and the cost of multiple measures across time. Instead, the analysis of emotional dimension of language and words used in the web and in social chats allows non-invasive multiple measures across time of affective condition of a population and represents an indirect but useful marker of psychiatric sufferance and mental distress.

Nevertheless, methodological limitations of our study must be acknowledged. First, we used Google Trends for the selection of our stimuli, but it only captures the search behavior of people who use Google and other search engines were thus excluded from this investigation. Second, our study employed an online task limited to Italian participants only. Consequently, we are not able to exclude that people from different nations and cultures or from a different social status (without Internet) might have be impacted differently by COVID-19. Third, we focused on two self-report measures and did not employ a multidimensional approach. More research is thus necessary to see if our initial findings replicate on people with different cultures and languages, socioeconomic status and with a multidimensional approach.

### Funding

This work was funded by a grant from MIUR (Dipartimenti di Eccellenza DM 11/05/2017 n.262) to the Department of General Psychology, University of Padova; Maria Montefinese was funded by a grant (Assegno di Ricerca) of the Department of General Psychology; furthermore, this study was supported by a free donation from the "Associazione poeti della parrocchia di Lama Polesine" to Alessandro Angrilli for studying the psychological impact of Covid-19. The funders had no role in study design, data collection and analysis, decision to publish, or preparation of the manuscript.

### Grant Disclosures

The following grant information was disclosed by the authors:
MIUR (Dipartimenti di Eccellenza DM): 11/05/2017 n.262.
(Assegno di Ricerca) of the Department of General Psychology.
Associazione poeti della parrocchia di Lama Polesine.

### Competing Interests

The authors declare there are no competing interests.

### Author Contributions

- Maria Montefinese conceived and designed the experiments, performed the experiments, analyzed the data, authored or reviewed drafts of the paper, and approved the final draft.
- Ettore Ambrosini conceived and designed the experiments, performed the experiments, analyzed the data, prepared figures and/or tables, authored or reviewed drafts of the paper, and approved the final draft.
- Alessandro Angrilli conceived and designed the experiments, performed the experiments, authored or reviewed drafts of the paper, and approved the final draft.

### Human Ethics

The following information was supplied relating to ethical approvals (i.e., approving body and any reference numbers):

The project has been approved by the Ethical Committee for the Psychological Research of the University of Padova (approved protocol reference number: 3563).

### Data Availability

All data and materials are available at Open Science Framework: Montefinese, Maria, Ettore Ambrosini, and Alessandro Angrilli. 2021. "Online Search Trends and Word-Related Emotional Response during COVID-19 Lockdown in Italy." OSF. June 5. doi: 10.17605/OSF.IO/GVH2Q.

### Supplemental Information

Supplemental information for this article can be found online at http://dx.doi.org/10.7717/peerj.11858#supplemental-information.

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
