# Peer review of "Online search trends and word-related emotional response during COVID-19 lockdown in Italy: a cross-sectional online study"

_PeerJ, doi:10.7717/peerj.11858_

## Round 0.1 · original submission · Major Revisions

Your manuscript was revised by 2 independent reviewers. Though they had a positive view of your manuscript, both raised questions about the clarity of the text and also the use of jargon that is too specific for those who work with psycholinguistics.

·

Basic reporting

The authors produced an interesting analysis on the language adopted by Italian people during their first lockdown through Google searches in combination with self-reported metrics of emotional well-being. Through a detailed and careful statistical analysis, the authors found that during the lockdown online searches on Google from Italian people shifted to higher frequency, more concrete, more positive and less arousing concepts. During confinement, individuals reported emotional states indicating a general loss of sense of agency and potential fearful states. Interestingly fear modulated different emotional dynamics in individuals. Those less subject to fear perceived concepts with less arousal, with additional losses of valence and dominance only in relation to COVID-related terms. Those people more prone to fearful states, ended up perceiving with less valence and dominance not only COVID-related but also COVID-unrelated concepts, a sign that the Italian lockdown provided enhanced emotional damage to the wellbeing of those people more sensitive to fear or other negative emotional states.

The manuscript is very well written, concise and clear. The topic is definitely of crucial relevance, given the need and the difficulty to quantify the emotional repercussions of lockdowns over individuals’ wellbeing. The authors have to be praised for their multidisciplinary approach, declining statistical techniques frequently used in psycholinguistics to a more social- and complexity-related scenario like the global pandemic of COVID-19. For these reasons I recommend the publication of this manuscript with minor revisions.

Experimental design

Lines 286 – 311: The report of the statistical testing is correct but it might confuse readers. Would it be possible to translate the information in text into a concise table to include in the main text? At the end of this section it would be useful to add a paragraph summarising the above tests in plain language, in order to better highlight the dynamics of affective ratings under the lockdown.

Validity of the findings

The Discussion could include a few more links and ties of the authors’ interesting results and those of the ever-increasing literature on COVID-19 and language analyses. I reported 3 main relationships that might be of relevance for the authors’ work:
- Dyer and Kolic (APN, 2020) found that online media indicated psychophysical numbing in people subject to restrictive health measures across 12 countries. Social discourse results being decreasingly emotional and increasingly analytic when discussing COVID-related jargon. This change detected in the language used by individuals could be interestingly connected with the emotional alterations identified by the authors, particularly in those individuals not being strongly exposed to maladaptation.
- Mazzuca et al. (PsyArxiv/4ndb8, 2021) found that the first Italian lockdown altered drastically the conceptual representation of concepts related to COVID-19, boosting the occurrence of conceptual associations from the illness domain like “flu-fear” or “disease-mask”. Could these shifts in the mental representation of the concepts be reconciled with the finding reported by the authors in this analysis? This connection would be interesting also in view of the authors’ analysis on how different individuals responded with different changes in their perceived emotional features of concepts.
- Stella (First Monday, 2020) investigated the language produced by Italians at the end of the first lockdown, in view of national re-opening. The analysis found that Italians tended to share more messages including a higher content in fear. The act of resharing is not always an endorsement but it is strongly triggered by a user sharing a similar view or an interest in the reshared content. In this way, the authors’ finding of improved feelings of fear and reduced sense of agency might be the source for the behavioural pattern found online in this paper although more research would be necessary in that direction.
Figure 1 – It seems to me like “hotel” was chosen as an example word for statistical reasons. However, the decrease of search occurrences for “hotel” is key to a more limited mobility being suddenly imposed to Italians. It might be worth mentioning this in the main text, especially since “hotel” are now being increasingly associated with COVID-19 (COVID-19 hotels are structures where quarantine or asymptomatic individuals can spend some time).

Reviewer 2 ·

Basic reporting

Overal the writing is relevant, concise and clear. The literature and introduction give a good description of the actual context of COVID-19 in Italy and insighful discussion of how this can be approached using classical approaches in emotion theory. Most of my suggestions are fairly minor. A list of presented below.

The results in the abstract (line 32-35) are compressed, making it difficult to parse without re-reading. “Lower valence and dominance judgments were given only to COVID-19-related words by individuals with less negative feelings and COVID-19 fear, but also to COVID-19-unrelated words by individuals with more negative feelings and COVID-19 fear..”

Instead, the following is clearer (verify that this interpretation is correct!)
“All individuals judged COVID-19 related words to be less positive and dominant. However, individuals with more negative feelings and COVID-19 fear also judged COVID-19 unrelated words to be less positive and dominant.”

“Moreover, arousal judgments for all words decreased and increased, respectively, for individuals with less and more negative feelings and COVID-19 fear.”

This could become “Arousal ratings increased for all words among individuals with more negative feelings and COVID-19 fear, but decrease among individuals with less negative feelings and COVID-19 fear.

Line 59. Start new paragraph for “The dimension theory of emotions.

Line 85. “To soften the influence of the COVID-19 emergency on individuals’ mental health, it is thus compelling the need to evaluate their emotional state in response to COVID-19 pandemic.” Please edit to make this less wordy.

Line 86. “Indeed, it has been reported that the COVID-19 affected the content explored in internet by people (Effenberger et al., 2020), as internet is an essential channel where people convey their feelings and emotions and seek health-related information (Kalichman et al., 2003; Reeves, 2001)”

Suggest to change as follows:
“Indeed, it has been reported that the COVID-19 affected the content that people explored online (Effenberger et al., 2020), and online media and platforms offer an essential channels where people convey their feelings and emotions and seek health-related information (Kalichman et al., 2003; Reeves, 2001)”

Line 95. Can you more explicitly draw out the relation between online searches and emotional reaction you plan to study? It's clear that online searches and emotional reaction will change due to COVID, but it's not clear how both research questions inform each other.

Line 104. “Findings will inform about the real-time estimation of the COVID-19 pandemic impact on
participants’ emotional response”. This is perhaps a bit too strong. Change to “Findings might inform about the real-time estimation...”

Line 121. Describe the nature of the RSV data in more detail. Was there a datapoint for each day?

Line 126: “RSV data for one word were not available.” Please indicate which word.

Line 155 -1 158. “This number of stimul was the largest that can be reliably rated during a single online session” I’m not sure what this means. Can you please explain? Many online studies involve a much larger set of stimuli.

Line 158 – 165. Is it possible to be more concrete? It seems that in this case the correlation is used a a measure of similarity and framing it in this manner might make it more clear.

Line 174. “…and for exemplar stimuli” and four exemplar stimuli

Line 210 Consider adding a few sentences to signpost what the structure of the data analysis is to improve readability.

Line 244. Please add a table with simple descriptives (mean, sd,n) for the data collected in the study (scales, word ratings) supplemented with the descriptives for the same set of words from the Italian ANEW.

Line 186: “… basilar description” → “basic description”

Line 198. “… from May 4 to May 17” Perhaps add “2020” since there have been additional lockdowns since.

Line 280: Consider adding a table that lists the significant words with largest differences for valence, arousal and dominance.

Line 350: “Figure 2G”. Should this be Figure 2?

Line 362. Affected means “were related” Change accordingly.

Experimental design

Line 119. Can you explain why the first four months of 2020 were chosen? Presumably no cases were found until February 21, so the preceeding period might be included as a baseline, but this wasn't clear at this stage.

Validity of the findings

Overal, I found the statistical analysis convincing, and commend the authors for taking a multiverse approach. I also found the power analysis useful.

The abstract mentions that the study involved healthy participants. How do you know whether the participants are healthy? Was this part of the questionnaire?

Line 155. Given the three measures, did they all provide te same outcomes?

Line 148. Can you indicate to what degree non-normality affected the data and whether a transformation reduced this issue (most likely skew)?

Line 237. In terms of analysis, it’s not clear what variables should be regarded as dependent and independent. In particular, treating PANAS as a DV would allow you a direct relation with fear, as a combination of (negative) valence and (high) arousal by adding those as interacting terms.

Additional comments

This work provides a nice demonstration of how emotional connotations of words is modulated by the emotional state of a person. It might be worth connection the findings reported here with previous literature that has manipulated the emotional state of participants who rate valence/arousal/dominance of words. Still, demonstrating this effect "in the wild" is something that is timely, especially in the context of COVID-19. Nice work!

---

## Round 0.2 · accepted · Accept

Your manuscript has been re-reviewed by one of the original reviewers and they have recommended its publication.

·

Basic reporting

The authors addressed all my points and considerably improved their final work. I recommend publication of the revised manuscript.

Experimental design

The authors addressed all my points and considerably improved their final work. I recommend publication of the revised manuscript.

Validity of the findings

The authors addressed all my points and considerably improved their final work. I recommend publication of the revised manuscript.

Additional comments

The authors addressed all my points and considerably improved their final work.

I recommend publication of the revised manuscript.